# A Trilayer Dressing with Self-Pumping and pH Monitoring Properties for Promoting Abdominal Wall Defect Repair

**DOI:** 10.3390/nano12162802

**Published:** 2022-08-15

**Authors:** Jie Hu, Guopu Chen, Gefei Wang

**Affiliations:** 1Department of General Surgery, Jinling Hospital, The First School of Clinical Medicine, Southern Medical University, Guangzhou 510515, China; 2Research Institute of General Surgery, Jinling Hospital, Medical School of Nanjing University, Nanjing 210002, China

**Keywords:** electrospinning, membrane, abdominal wall defect, pH monitoring, directional biofluid transport, wound healing

## Abstract

Due to abdominal infection, excessive wound exudation, and intestinal fistula formation, the treatment of full-thickness abdominal wall defects has become a difficult challenge for clinic doctors. This clinical problem cannot be resolved with existing biomaterials. To facilitate the repair of the abdominal wall, we developed a novel wound dressing with directional biofluid transport. We used electrospinning to spin a trilayer dressing consisting of hydrolyzed poly-acrylonitrile (HPAN)/Curcumin (CUR), polyurethane (PU), and polycaprolactone (PCL). In vitro results show that the three-layer wound dressing is biocompatible, capable of directional transport of excessive wound exudation, preventing reverse penetration, and monitoring the pH of the wound. Furthermore, in vivo results show the trilayer wound dressing improves the wound microenvironment, reduces inflammatory factors, promotes angiogenesis, and accelerates abdominal wall repair. Thus, we believe that the novel trilayer electrospinning dressing could facilitate abdominal wall defect repair.

## 1. Introduction

Full-thickness abdominal wall defect is accompanied by numerous complications, including intra-abdominal infection, intestinal fistula, and water-electrolyte imbalance, making wound protection and repair a challenging problem for clinicians [1,2,3]. Due to enormous tension and infection, the wound surface cannot be directly reconstructed [4]. Typically, commercial polypropylene (PP) mesh is used to temporarily close the abdominal wall. Meanwhile, Excessive wound exudate leads to bacterial proliferation and disturbance of the wound microenvironment. Negative pressure wound therapy (NPWT) is employed to reduce excess exudate [5,6]. However, PP mesh is susceptible to foreign body reaction, leading to the development of intestinal fistula due to rubbing of the intestinal surface and lack of biological functions, and the polyurethane sponge of the negative suction device also easily induces intestinal fistula [7]. Additionally, the pH of a wound can indicate the presence of an infection. The normal pH range for skin is 5–7, which is weakly acidic. Following an injury, the pH value of the normal skin is rendered alkaline. Subsequently, the pH value decreases and becomes acidic following an infection [8,9]. Therefore, it is imperative to create a novel dressing with a pH monitoring property to protect the exposed bowels and pump excess exudate to speed up abdominal wall repair.

Electrospun membranes are widely used in biological dressings, drug delivery, tissue engineering scaffolds, etc., due to their excellent air permeability, biocompatibility, and mechanical properties [10,11,12,13,14,15]. The traditional hydrophilic electrospun dressing can absorb exudate, but the exudate remains on the wound surface [16]. Accordingly, directional water transport electrospinning fiber membranes have developed rapidly in recent years, which are composed of a non-wetting nanofiber membrane [17,18,19,20,21]. Owing to the difference in wettability of the fiber membrane, the directional water transport electrospinning fiber membrane can transport water from the hydrophobic layer to the hydrophilic layer and is widely used in cleaning, biomedicine, textile, and other fields [22,23,24,25,26,27]. However, most composite membranes are bilayer structures, and their weak water transport prevents reverse osmosis capacity, limiting their applications [28]. Hence, a trilayer electrospinning membrane with enhanced water pumping and prevention of osmosis in reverse capabilities is anticipated in protecting abdominal wall defect open wounds and expediting abdominal wall repair.

The biocompatible wound dressing offering directional biofluid transport was composed of a hydrolyzed polyacrylonitrile (HPAN)/curcumin (CUR), polyurethane (PU), and polycaprolactone (PCL) electrospinning membrane (Figure 1). HPAN/CUR electrospun membranes were utilized as the surface layer for transporting wound exudate and monitoring wound pH. The hydrolysis of PAN yields, whose hydrophilicity is enhanced by the conversion of a large number of cyano groups to carboxyl groups in an alkaline environment [23,29]. Curcumin (CUR) exhibits different colors at varying pH levels and has been widely utilized as a pH indicator [30,31,32,33,34,35]. Additionally, polyurethane (PU) was selected as the second layer to absorb water and promote its migration to the surface layer. Moreover, hydrophobic polycaprolactone (PCL) was chosen as the inner layer to prevent water reflux. The in vitro tests demonstrated low immediate cytotoxicity and excellent hemocompatibility, whereas the animal studies demonstrated that the dressing effectively promoted abdominal wall repair by draining excess exudate, improving the wound microenvironment, reducing wound inflammation, and accelerating angiogenesis. Therefore, we believe that this trilayer electrospinning dressing is promising for the treatment of abdominal wall defects.

## 2. Materials and Methods

### 2.1. Materials

Polycaprolactone (Mn 60,000–65,000, PCL) and Polyacrylonitrile (Mw 150,000, PAN) were purchased from Shanghai Macklin Biochemical Co., Ltd. (Shanghai, China). Polyurethane (PU) was purchased from Huafon Co. Ltd. (Zhejiang, China). 2,2,2-Trifluoroethanol, N, N-Dimethylformamide (DMF), Chitosan (CS), and Curcumin (CUR) were purchased from Aladdin Reagent Co., Ltd. (Shanghai, China). Mouse L929 cells were purchased from Procell Co., Ltd. (Taiwan, China). DMEM and Live/Dead viability assay kit (KGAF001) for animal cells was purchased from Keygen Biotechnology Co. Ltd. (Jiangsu, China). All male SD rats (200 g) were provided by Jinling Hospital. The rats unrestrictedly obtained food and water and were treated with the “Guidelines for the Care and Use of Laboratory Animals”. All of the experiments and animal were approved by the Animal Investigation Ethics Committee of Jinling Hospital.

### 2.2. Preparation of Electrospun Membranes

PCL was dissolved in 2,2,2-Trifluoroethanol to produce a 10 wt.% PCL solution, and the solution was stirred at 26 °C for 6 h. The dissolution of PU solids in DMF solution yielded a 15 wt.% PU solution. PAN and CUR were dissolved in a DMF solution to 15 wt.% PAN and 8% CUR. The PCL solution was used to electrospun with a 24 G needle at 15 kV, using an electrospinning device (JNS-MS-05-ME01, Nanjing Janus New-Materials Co., Ltd. (Jiangsu, China) at 15 kV. The distance between the collectors and the needle tip was 15 cm. Subsequently, the PU and PAN/CUR spinning solution were sequentially used to carry out electrospinning, as previously described. The fiber membrane was then soaked in an alkali solution for 5 min to convert PAN to HPAN. The fiber membrane was then soaked in deionized water to remove the alkali solution and vacuum-dried to remove any residual solvent.

### 2.3. Morphology of Electrospun Membranes

The morphology of PCL, PU, and HPAN/CUR were characterized using scanning electron microscopy (SEM, Regulus8100, Hitachi Co., Ltd., Japan). Prior to the imaging, the fibrous membranes were coated with gold and the samples were examined at an accelerating voltage of 20 kV. Additionally, Image J software was used for image analysis to determine the diameters of fibers and porosities of electrospinning.

### 2.4. Water Contact Angle of Electrospun Membranes

Briefly, a 3 μL water was dropped on the membrane’s top face, and the CA values at various times were measured by WCA meter (OCA20, Dataphysics Co., Ltd., Filderstadt, Germany).

### 2.5. pH Indicator Test of Fibrous Membranes

200 μL PBS solutions with different pH values were dropped on the HPAN/CUR nanofibers. The color of the HPAN/CUR nanofibers treated at different pH and the CUR solution at different pH were subsequently photographed. The photographs were then analyzed using Adobe Photoshop. Additionally, we measured the absorbance of CUR solution using a UV spectrophotometer.

### 2.6. Draining Ability and Prevention of Water Reversed Penetration

The ink was utilized as an exudate to test the biofluid draining capability of electrospun membranes in the abdominal wall defect wound. To test the draining ability of the hydrophobic bilayer and trilayer electrospun membranes, 200 µL of blue ink was dropped onto the hydrophobic surfaces and photographs were taken at different times. On the hydrophilic of bilayer electrospun membranes and trilayer electrospun membranes, 200 L of blue ink was dropped to test the prevention of water reversed penetration, and the residual ink on the bottom paper was photographed.

### 2.7. Cytotoxicity Test and Hemolysis Test

For the cytotoxicity test, the blank group (no added materials) was used as the control group, and the PCL, bilayer (PCL/PU), and trilayer were used as three experimental groups. These different samples were soaked in DMEM for 24 h after being UV sterilized. The L929 cells were subsequently treated with a leaching solution [20]. Additionally, the CCK-8 kit was used to determine the viability of L929 cells. A Live-Dead Staining Kit was utilized to stain the seeded cells after 24 h and 48 h. Those photographs were then photographed by a fluorescence microscope.

The hemolysis test was used to evaluate the blood compatibility of the electrospun film. Red blood cells were obtained by centrifuging the whole blood at 1000 rpm for 8 min at 4 °C and washing it using a normal saline solution. Various 1 mg samples were soaked in normal saline solutions containing 5% blood cells. DI water was used as a positive control and the normal saline solution was used as a negative control. After 2 h, the solutions were centrifuged to collect the supernatant, and the absorption was measured at 540 nm. The hemolysis rates were computed using the following equation:Hemolysis rate (%) = (OD sample/OD positive group) × 100%

### 2.8. In Vivo Full Abdominal Wall Defect Study

To test the effect of electrospun membrane on abdominal defects, 9 SD rats (200–250 g) were used. We incised a hole (diameter: 2 cm) in the full-thickness abdominal wall of anesthetized rats. The control group was administered commercially available polypropylene mesh, while the experimental group was administered the bilayer and the trilayer electrospun membrane. We removed the dressing on day 7. Abdominal wall defects in rats were photographed on days 0, 5, and 14. On the 14th day, the new granulation tissue of the abdominal wall was collected for subsequent experiments. The wound healing process was simulated using one photograph of a representative animal wound in each group. The testing wound pH value was analyzed by color photograph and the actual pH value of the wound was tested with pH precision test paper on days 0, 1, 2, and 3.

### 2.9. Histology, Immunohistochemistry, and Immunofluorescence Staining

Tissue samples were dehydrated, paraffin-embedded, and microtome-sliced. For HE staining, hematoxylin-eosin was used. For immunohistochemical staining of IL-6 and TNF-α, IL-6 antibody (Abcam, ab9324) and anti-TNF-α antibody were employed (Abcam, ab109322). Anti-CD31 antibody (Abcam, ab64543) was used as the primary antibody to label blood vessels for immunofluorescence staining.

### 2.10. q-PCR

Tissue samples were soaked in Trizol to extract RNA. The mRNA was subsequently quantified using quantitative real-time PCR. The primers in q-PCR are listed in Appendix A.

### 2.11. Statistical Analyses

We used either a one-way ANOVA, two-way ANOVA test and *t*-test to analyze statistical differences. Quantitative data were indicated as mean ± standard deviation (SD). Additionally, differences at *p* < 0.05 (* < 0.05, ** < 0.01, *** < 0.001) were considered significant.

## 3. Results and Discussions

### 3.1. Characterization of Electrospinning Membrane

Using an electrospinning machine and a layer-by-layer assembly technique, we electrospun this dressing. All electrospinning membranes in Figure 1A have a uniform fiber thickness, a smooth surface, and a random directional distribution. Additionally, the PU membranes have the thinnest fiber diameters compared to other groups, whereas the PCL membranes have the thickest fiber diameters. The pore sizes of the three membranes are also similar, ranging from a few microns to tens of microns. The three-layer structure can also be observed in the cross-sectional SEM image (Figure 1B). As shown in Figure 1C–E, the diameters of HPAN/CUR, PU and PCL fibers were 1228.0 ± 231.5, 462.7 ± 119.8, and 1930.0 ± 447.1 nm, respectively. The porosities of HPAN/CUR, PU, and PCL were 43.4 ± 3.2%, 56.7 ± 7.0%, and 55.0 ± 6.9%, respectively (Figure 1F), indicating that water can be directionally transported through the fiber membrane pores. The dressing must be sufficiently flexible to conform to irregular wound surfaces. Accordingly, the electrospun dressing can be easily bent and folded into multiple angles, demonstrating a high degree of flexibility (Figure 1G). Water directional transport depends on the difference between the hydrophilicity and hydrophobicity of fiber membranes; therefore, we measured the water contact angle of each fiber membrane. In Figure 2A, the water contact angle of PCL is 123.3 ± 2.1° in 1 s. The water contact angle of PCL does not change much over time and is 103.0 ± 0.5° at 240 s. This suggests that PCL possesses certain hydrophobic properties and can serve as the bottom layer to prevent reverse osmosis of exudate. PU is more hydrophilic than PCL, with a water contact angle of 101.7 ± 0.9° at 1 s. As PU has been modified, it undergoes significant changes when exposed to water, becoming 81.0 ± 0.5° after 60 s and 14.2 ± 0.7° after 240 s. This property of the material enables the PU membrane to function as an intermediate transition layer. Even with the addition of a small amount of hydrophobic cur, HPAN/CUR remains hydrophilic because the hydrolysis of PAN generates a large number of carboxyl groups. Accordingly, the water contact angle of HPAN/CUR is 64.7 ± 0.9°, 46.6 ± 0.2°, 39.5 ± 0.3°, and 26.5 ± 0.3° at 1, 2, 3, and 4 s, respectively. The water droplets are quickly absorbed at the drop of 8 s, which adequately demonstrates the excellent water absorption capacity of HPAN/CUR.

### 3.2. pH Response of Electrospinning Membrane

Curcumin is an FDA-approved natural pigment that exhibits different colors at different pH levels. We evaluated 8% CUR solutions absorbance with varying pH values (Figure 2C). Accordingly, the UV-Vis spectrum shows different absorption peaks at pH 5.5, 6.5–7.5, and 8.5. Subsequently, image color for the L, A, and B channels of the CUR solutions and trilayer electrospun membrane were analyzed at pH 5.5, 6.5, 7.5, and 8.5. The color and LAB values are distinct at pH 5.5, 6.5, 7.5, and 8.5, indicating that the trilayer electrospun membrane could serve a pH indicator in the wound healing process. As pH increased, the color of the 8% CUR solution transformed from the pale yellow to the orange. However, in an acidic environment, the color of the solution hardly changes, but when it becomes alkaline, the color changes from yellow to orange (Figure 2D,E).

### 3.3. Directional Water Transport Capability and Reverse Permeation Prevention

To test the water transport ability of the dressing, 200 μL of blue ink was deposited on the hydrophobic side. As evident in Figure 3A, both the bilayer and trilayer electrospun membranes have the capacity to pump water, with the trilayer membrane having a greater capacity than the bilayer membrane. Additionally, the blue ink was absorbed rapidly and absorbed completely in 240 s in the trilayer electrospinning membrane group, while the complete absorption of the blue ink in the bilayer group required 300 s (Figure 3C). Moreover, we dropped ink on the hydrophilic side of the trilayer membrane, which revealed that the water was quickly absorbed by the HPAN/CUR within a few seconds. Additionally, no liquid was observed on the bottom of the paper, indicating that the PU layer made it difficult for the water to reach the PCL membrane. However, water infiltration was slow on the bilayer membrane, and blue ink appeared at the bottom after long-term infiltration, which could be attributed to the absence of a buffer layer, resulting in the water droplets directly contacting the PCL membrane after absorption and diffusion (Figure 3B). We also calculated the wetting area in Figure 3D, which demonstrated that the wetting area of the bilayer membrane is greater than that of the trilayer membrane. The potential mechanism for the directional transport of water by the electrospun film is presented in Figure 3E. As evident from the figure, a hydrophobic force is generated by the hydrophobic properties of the membrane, whereas a hydrostatic force is generated by the water’s gravitational pull. The capillary force can be used via the Young–Laplace equation (Laplace-pressure = 4γ*cosθ/Dpore), where γ, θ, and Dpore represent the surface tension, water contact angle, and the pore size, respectively [28]. This capillary force is directly proportional to the pore size of the film and inversely proportional to the water contact angle of the film. However, in practical applications, we have observed that the pore size has a negligible effect and that the water contact angle plays the most important role. When water droplets drip from the hydrophobic side, they are initially exposed to the hydrophobic effect of the PCL membrane and hydrostatic pressure. The water droplets then pass through the hydrophobic layer by the capillary force of PU and then transfer to the lowest layer by the capillary force of HPAN. Conversely, when a water droplet drips from the hydrophilic side, capillary force causes it to spread to both sides. Subsequently, when water droplets reach the hydrophobic layer, the capillary force and hydrophobic force of the upper layer are greater than the hydrostatic force, preventing them from penetrating downward.

### 3.4. Biocompatibility of the Trilayer Electrospinning Membrane

Because a wound dressing is in direct contact with the wound surface, it must have good biocompatibility. The biocompatibility of the trilayer membrane was determined through cytotoxicity and hemocompatibility tests. Figure 4A,B demonstrates that the number of dead cells is small in both the 24 h and 48 h groups, with the 48h group containing more dead cells than the 24 h group. Additionally, the live-dead staining confirmed the good biocompatibility of the PCL, bilayer, and trilayer membranes. Concurrently, we utilized CCK8 kit to determine cell viability (Figure 4C). At 24 h, the cell viability of all groups are greater than 90%, but decreased as the number of membrane layers increased. Similar to the results of the fluorescence photographs, the cell viability at 48 h was greater than that at 24 h, but the overall mortality rate was greater than 80%. Additionally, we evaluated blood compatibility, which revealed that the hemolysis rate of the fibrous membrane was less than 5%. This suggests that none of the membranes will cause hemolysis and have the potential for use as a biocompatible material (Figure 4D).

### 3.5. In Vivo Experiments

We also examined the effect of the electrospun membrane dressing on wounds using a rat model of full-thickness abdominal wall defect. We created a circular full-thickness defect (diameter: 2 cm) in the abdominal wall of rats, inserted PP mesh, bilayer electrospinning membrane, and trilayer electrospinning membrane into the defect, and took photographs on the fifth and fourteenth days to determine the wound area (Figure 5A–C). On the fifth day, the wounds in the PP mesh group were red, swollen, whitish, and full of exudations, whereas the wounds in the other two groups of dressings had no exudations and the wound area of the three-layer film was smaller than that of the double-layer film. It reflects the function of the electrospinning dressing to absorb the exudate, and the drainage effect of the trilayer is superior to that of the bilayer. At day 14, the wound area caused by the trilayer membrane was minimal and demonstrated the ability to promote abdominal wall repair. The pH of a wound is frequently associated with infection, and abdominal wall defects are frequently accompanied by abdominal infection; therefore, monitoring the pH of a wound can indicate abdominal infection. We measured the pH level of the wound and compared the color of the surface of the dressing (Figure 5D). On the first day of the wound, the pH was observed to decrease before increasing continuously. Further, the pH value indicated by the dressing was very close to the actual pH value, thus, it can be used as an indicator to monitor the wound infection.

To further investigate the effect of dressing, we performed a pathological analysis of the newly formed granulation tissue. In Figure 6A, the group treated with a trilayer dressing had the greatest thickness of abdominal wall granulation tissue, and the intestinal serosa layer remained intact, demonstrating the ability to promote abdominal wall healing and to protect the exposed intestinal surface. In the PP group, bilayer group, and trilayer group, the granulation tissue was 1.1 ± 0.1, 1.7 ± 0.1, and 2.3 ± 0.2 mm thick, respectively. We also detected the inflammatory factors IL-6 and TNF-α to determine the level of inflammation in the wound. The PP group had the highest level of inflammation, whereas the trilayer dressing group had the lowest level of inflammation, indicating that the dressing can drain excess exudate and reduce the level of inflammation. Due to the reduction of inflammation, the wound healing process could rapidly change from an inflammatory phase to a granulation phase, which stimulates angiogenesis and further accelerates wound healing. As anticipated, the trilayer dressing had the highest amount of angiogenesis. The densities of the blood vessels in the PP group, bilayer group, and trilayer group are 42.3 ± 2.5, 95.0 ± 13.2, and 120.0 ± 10.0/mm^2^, respectively (Figure 6C). The level of inflammatory factors and growth factors could reflect the situation of tissue growth [36,37]. We determined the mRNA levels of pro-inflammatory (IL-6, TNF-α) and pro-healing (IL-4, IL-10, TGF-β, PDGF). In Figure 6D, the expression levels of inflammation-related mRNA in the PP mesh group were higher than those in the trilayer dressing group, whereas the expression levels of healing-related mRNA were lower. Thus, the results successfully demonstrate the dressing’s ability to reduce inflammation and to promote wound repair. Although good curative effects have been observed in animal experiments, due to the limitations of faster healing in animal models and certain differences between humans, more clinical experiments should be conducted to explore the real curative effects. In addition, rats are highly resistant to infection and further studies could add antibacterial components to increase the functional versatility of the dressing.

## 4. Conclusions

We designed a trilayer electrospinning dressing with directional water transport for abdominal wall defect wound healing. The electrospun dressing demonstrated superior mechanical properties, high flexibility, and stability. Additionally, the excellent directional pumping capability of the electrospun trilayer dressing was verified and its ability to prevent reverse osmosis was confirmed. In addition, it also demonstrated a precise pH monitoring function. Moreover, the cytotoxicity and hemocompatibility of the dressing were validated from the results of the in vitro experiments. Additionally, the dressing was found to drain excess exudate, improve the wound microenvironment, reduce wound inflammation, accelerate angiogenesis, and promote abdominal wall repair in the animal studies conducted. Thus, the novel dressing designed based on clinical requirements has great potential, and we believe it can be utilized in abdominal wall defects.

## Data Availability

Not applicable.

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
