# Peer review of "A Trilayer Dressing with Self-Pumping and pH Monitoring Properties for Promoting Abdominal Wall Defect Repair"

_nanomaterials, 2022, doi:10.3390/nano12162802_

Round 1

Reviewer 1 Report

This paper describes new tri-layer electrospun device which is designed to absorb extrudate from a wound site. The overall concept is very interesting and the data suggests that it has promise. However data presented I cannot be sure of its significance, mainly the number of animals use in the study is not stated, many the figures lack enough description eg. Fig 5B what do the colors mean is this of one animal or an average, Fig 5 C, how many replicates and what are the error bars? (very frustrating as reviewer that is missed) Fig 5 D real pH, no description in methods how this was done and testing pH (what is this and what treatment). The abstract is not a summary of aim, methods, results (no data) but more introduction with a conclusion. The authors do not discussion any limitations to the study, how would this translate to humans, for example the rate of absorption was measure but not capacity, why was the capacity of absorption determined?, would human wound extrude the same amount as a small animal. What about the long term implications of the implanted material? Due to not disclosing the replication of the animal trial i cannot accept these findings and conclusions despite the in vitro looking promising...

Reviewer 2 Report

- Figure 1-B: it is unclear from the image how the three regions are identified in the cross-section. No distinct features seem to be detectable from this image.

- Figure 1C-F: what was the sample size for these quantitative data? 'n' should be reported for ALL quantitative data presented throughout the manuscript. Also, statistical analysis is missing for 1F.  Make sure to add stats for ALL quantitative data that need that.

- All acronyms must be defined in the figure captions. Also, the figure captions are all too brief and do not elaborate all the details related to each figure.

- Figure 2: 'LAB' must be defined.

- Figure 4: again, the caption is too short. Elaborate all the details. For instance, A and B are not clearly explained (what is the difference?). Control group must be defined. Also, cell viability is presented only for 2 days which is pretty short to evaluate the biocompatibility of these materials. This must be extended at least for one week.

- Figure 5 - the animal study: did the team use an injured group without any treatment? That control group is certainly needed to complete and validate the comparison across groups. Also, why is the wound pH data presented only for 3 days, while the rest are continued for 14 days? How does the pH change after the first few days? This needs to be consistent.

- Figure 6: again, the control group (injury without dressing) is needed. The IHC images (CD31) are pretty pale and hard to read. How was the vessel density measurement (panel C) done?

Round 2

Reviewer 1 Report

I am satisfied with the amendments made by authors 

Reviewer 2 Report

I believe the revised manuscript merits publication in Nanomaterials.